# Active Food Packaging Coatings Based on Hybrid Electrospun Gliadin Nanofibers Containing Ferulic Acid/Hydroxypropyl-Beta-Cyclodextrin Inclusion Complexes

**DOI:** 10.3390/nano8110919

**Published:** 2018-11-07

**Authors:** Niloufar Sharif, Mohammad-Taghi Golmakani, Mehrdad Niakousari, Seyed Mohammad Hashem Hosseini, Behrouz Ghorani, Amparo Lopez-Rubio

**Affiliations:** 1Department of Food Science and Technology, School of Agriculture, Shiraz University, km 12 Shiraz-Esfahan Highway, 71441-65186 Shiraz, Iran; sharif1986@shirazu.ac.ir (N.S.); golmakani@shirazu.ac.ir (M.-T.G.); niakosar@shirazu.ac.ir (M.N.); hhosseini@shirazu.ac.ir (S.M.H.H.); 2Department of Food Nanotechnology, Research Institute of Food Science and Technology (RIFST), km 12 Mashhad-Quchan Highway, 91895/157/356 Mashhad, Iran; b.ghorani@rifst.ac.ir; 3Food Quality and Preservation Department, IATA-CSIC, 46980 Paterna, Valencia, Spain

**Keywords:** gliadin, ferulic acid, hydroxypropyl-beta-cyclodextrin, electrospinning

## Abstract

In this work, hybrid gliadin electrospun fibers containing inclusion complexes of ferulic acid (FA) with hydroxypropyl-beta-cyclodextrins (FA/HP-β-CD-IC) were prepared as a strategy to increase the stability and solubility of the antioxidant FA. Inclusion complex formation between FA and HP-β-CD was confirmed by Fourier transform infrared spectroscopy (FTIR), differential scanning calorimeter (DSC), and X-ray diffraction (XRD). After adjusting the electrospinning conditions, beaded-free fibers of gliadin incorporating FA/HP-β-CD-IC with average fiber diameters ranging from 269.91 ± 73.53 to 271.68 ± 72.76 nm were obtained. Control gliadin fibers containing free FA were also produced for comparison purposes. The incorporation of FA within the cyclodextrin molecules resulted in increased thermal stability of the antioxidant compound. Moreover, formation of the inclusion complexes also enhanced the FA photostability, as after exposing the electrospun fibers to UV light during 60 min, photodegradation of the compound was reduced in more than 30%. Moreover, a slower degradation rate was also observed when compared to the fibers containing the free FA. Results from the release into two food simulants (ethanol 10% and acetic acid 3%) and PBS also demonstrated that the formation of the inclusion complexes successfully resulted in improved solubility, as reflected from the faster and greater release of the compounds in the three assayed media. Moreover, in both types of hybrid fibers, the antioxidant capacity of FA was kept, thus confirming the suitability of electrospinning for the encapsulation of sensitive compounds, giving raise to nanostructures with potential as active packaging structures or delivery systems of use in pharmaceutical or biomedical applications.

## 1. Introduction

Ferulic acid (FA, 4-hydroxy-3-methoxy cinnamic acid), a phenolic compound classified in the group of the hydroxycinnamic acids, is present in commelinid plants such as rice, wheat, oats, and some vegetables, fruits, and nuts [1]. FA exhibits a wide range of biological and biomedical effects including antioxidant [2], anti-inflammatory [3], anti-diabetic [4], hepatoprotective [5] and anti-carcinogenic [6], among others. In addition, FA has been approved as a food additive, hindering the peroxidation of lipids due to its scavenging capability of superoxide anion radicals [7,8]. Despite its biological activity, direct incorporation of FA to foods is limited due to its low water solubility [9] and poor stability under physical and thermal stresses, highlighting its susceptibility to light and oxygen exposure [10]. Interestingly, these limitations might be overcome by the formation of inclusion complexes with cyclodextrins (CDs).

CDs are cyclic, natural, and nontoxic oligosaccharides produced by the enzymatic degradation of starch [11]. Based on the number of α-1,4-linked glucopyranose units in their cyclic structure, different CDs exist, being the most common ones α-CD, β-CD, and γ-CD having 6, 7, and 8 glucopyranose units, respectively [12]. CDs have toroid-shaped molecular structures, a hydrophobic internal cavity and a hydrophilic external surface that make them capable of forming noncovalent host−guest inclusion complexes with a variety of molecules such as phenolic compounds [13]. Therefore, this unique capability offers outstanding improvements in the properties of the guest molecules including protection from degradation and oxidation, enhancing solubility, chemical stability and controlling the release rate [14,15].

Recently, electrospinning, a versatile and cost-effective technique has gained a great interest for fiber fabrication in the range of micron, submicron, and nanoscales [16]. The electrospun fibers have key advantages including high surface-to-volume ratios, small pore sizes and high porosity [17,18]. From an application standpoint, biopolymer-based electrospun fibers can be used for bioactive encapsulation with potential use as delivery systems, controlled release agents or active packaging structures, amongst others [19]. The combination of biopolymers as fiber matrices with organic and inorganic particles (such as clay nanoparticles) for the release of functional molecules has been shown to be an excellent strategy to produce bioactive packaging structures [20,21,22,23]. Amongst the natural biopolymers to be used as electrospun matrices, proteins have been widely studied due to their renewable, biodegradable, and biocompatible character [24]. Particularly prolamins, which are plant storage proteins normally obtained as by-products of the starch or beta-glucan production processes, are interesting raw materials for fabricating cost-effective electrospun structures [25], as they are considered more “environmentally economical” when compared with proteins from animal sources [26]. Gliadins, storage prolamins present in wheat kernels, consists of a central domain containing highly repetitive amino acid sequences including proline and glutamine residues and hydrophobic terminal domains which surround the central part. Therefore, gliadin has amphiphilic properties [27]. Gliadins are poorly soluble in aqueous solutions except at extreme pH conditions [28]. It has been found that gliadin has bioadhesive properties, being able to interact with the intestinal membrane through electrostatic interactions and hydrogen bonding [29]. Therefore, gliadins are ideal candidates to fabricate electrospun fiber mats for various applications.

Given the potential of CDs to increase the stability and solubility of hydrophobic bioactives and the excellent properties of gliadin as encapsulation matrix, the aim of the present work was to combine both structures to generate hybrid systems to be used as active coatings for the protection and enhanced solubility of FA within food products. To this end, solid FA/hydroxypropyl-Beta-cyclodextrin inclusion complexes (FA/HP-β-CD-ICs) were first formed, which were subsequently incorporated within gliadin fibers through electrospinning. The inclusion complexes were characterized by X-ray diffraction (XRD), differential scanning calorimeter (DSC), thermogravimetric analysis (TGA) and Fourier transform infrared (FTIR) spectroscopy. Gliadin fibers containing free FA were produced as control samples. The morphological characterization of the hybrid fibers was carried out by scanning electron microscope (SEM). In addition, the presence and distribution of FA in HP-β-CD-ICs and fibers were investigated by fluoresce microscopy. The thermal stability, photostability, release behavior and antioxidant capacity of the developed fibers were also evaluated.

## 2. Materials and Methods

### 2.1. Materials

Wheat gluten was purchased from a local shop (Shiraz, Iran). Ferulic acid (Carbosynth, Newbury, England), hydroxypropyl-β-cyclodextrin (Carbosynth, Newbury, England), 2,2-Diphenyl-1-picrylhydrazyl (DPPH, Sigma-Aldrich, St. Louis, Missouri, United States), phosphate buffer solution (PBS, pH = 7.3 ± 0.1, DNAbiothec, Tehran, Iran), acetic acid (Scharlab, Barcelona, Spain) and ethanol (Panreac, Barcelona, Spain) were purchased and used as received without any further purification. All other chemicals used were of analytical grade unless otherwise specified.

### 2.2. Preparation of Solid FA/HP-β-CD-IC

The inclusion complexes between FA and HP-β-CD were prepared using the freeze-drying method described by Kfoury, Auezova [30]. FA and HP-β-CD were mixed in aqueous solution in a 1:1 M ratio at a concentration of 10 mM, mixing for 24 h at room temperature. Then, the solution was filtered, frozen and lyophilized by a laboratory freeze dryer (ALPHA 2-4 LD plus, Martin Christ, Osterode am Harz, Germany) at 85 °C and Pa for 48 h. The inclusion ratio (IR%) was calculated using Equation (1):IR%= (experimental FA content in the solid IC/theoretical FA content) × 100(1)

### 2.3. Gliadin Extraction

The gliadin fraction of gluten was extracted using the method described by Hong, Trujillo [31] with slight modifications. Briefly, samples of dried gluten powder (20 g) were gently stirred in an ethanol/water mixture (70/30 *v*/*v*; gluten/solvent ratio of 1/12) for 4 h at 20 °C. The suspension was centrifuged to collect the gliadin fraction at 10,000 g for 10 min. Finally, the ethanol was evaporated at ambient conditions. The extraction yield of gliadin from wheat gluten powder was 37.5%. The protein content as determined using the Kjeldahl method was 89.8% on a dry matter basis.

### 2.4. Preparation and Characterization of Gliadin Solutions for Electrospinning

Initially, 25% (*w*/*v*) gliadin solutions were prepared by stirring the protein powder in acetic acid at ambient conditions until complete dissolution. Subsequently, FA or FA/HP-β-CD-IC were added into the gliadin solutions (at concentrations of 5, 10 and 20% *w*/*w*, with respect to the biopolymer).

The solution properties that affect the electrospinning process, specifically the apparent viscosity, surface tension, and electrical conductivity were evaluated. The surface tension of the solutions was measured using the Wilhemy plate method in an EasyDyne K20 tensiometer (Krüss GmbH, Hamburg, Germany) after calibration of the equipment with deionized water. The electrical conductivity of the solutions was measured using a conductivity meter XS Con6 (Labbox, Barcelona, Spain). The apparent viscosity of the gliadin solutions was determined by a rotational viscometer VISCO BASIC PLUS L from Fungilab S.A. (Sant Feliu de Llobregat, Spain) at 10 rpm using the TL5 spindle. All measurements were made at 25 °C. All Experiments were performed, at least, in triplicate.

### 2.5. Hybrid Gliadin-Based Fiber Formation Through Electrospinning

Gliadin fibers incorporating free FA (G-FA) and FA/HP-β-CD-IC (G-FA/HP-β-CD-IC) were fabricated via electrospinning. The electrospinning process was conducted using an electrohydrodynamic apparatus equipped with a variable high voltage 0–35 kV power supplier (spinner-3X-Advance, ANSTCO, Tehran, Iran). Solutions were loaded into 10 mL disposable plastic syringes and the electrospinning process was conducted at the voltage of 18 kV and a flow rate of 1 mL/h. Tip to collector distance was kept constant at 100 mm. The obtained fibers were collected on aluminum foil attached to the surface of the collector and kept overnight under the hood to evaporate any solvent residues. All experiments were performed at ambient conditions.

### 2.6. Ultraviolet-Visible Spectroscopy

The Ultraviolet-Visible (UV-Vis) spectra of FA, HP-β-CD and their corresponding inclusion complexes were recorded on a UV-Vis spectrophotometer (UV-1280, Shimadzu Corporation, Kyoto, Japan). Each sample (0.3 mM) was dissolved -in methanol and the spectra were obtained in the range from 220 to 400 nm.

### 2.7. Optical and Scanning Electron Microscopy (SEM)

The morphology of electrospun gliadin structures containing FA and FA/HP-β-CD-IC was examined by a TESCAN-Vega 3 scanning electron microscope (SEM) (TESCAN, Brno, Czech Republic). SEM was conducted at an accelerating voltage of 20 kV and at working distances of 9-16 mm after sputter coating the electrospun webs with gold under vacuum (Q 150R-ES; Quorum Technologies, Laughton, UK). Image analysis software (Digimizer, MedCalc Software, Ostend, Belgium) was used to determine fiber diameters from the SEM micrographs in their original magnification. Average fiber diameters (AFD) and fiber size distributions were obtained from a minimum of 100 measurements. The presence and distribution of FA in HP-β-CD inclusion complex and fibers were investigated using a digital microscopy system (Nikon Eclipse 90i, Barcelona, Spain) fitted with a 12 V, 100 W halogen lamp and equipped with a digital imaging head which integrates an epifluorescence illuminator. A digital camera head (Nikon DS-5Mc, Tokyo, Japan) with a 5-megapixel CCD cooled with a Peltier mechanism was attached to the microscope.

### 2.8. Encapsulation Efficiency

Encapsulation efficiency (EE%) was calculated by measuring the non-entrapped FA according to Yang, Feng [32] with some modifications. Briefly, Fibers (10–20 mg) was submerged in absolute ethanol (8 mL) for 30 s. Then the mixture was centrifuged at 2500 rpm for 10 min and the absorbance of FA was then determined by UV-Vis spectrophotometer (UV-1280, Shimadzu Corporation, Kyoto, Japan) based on the calibration curve (R^2^ = 0.999) obtained for FA in absolute ethanol at a wavelength of 310 nm. The EE% values were calculated using Equation (2):EE% = ((total theoretical mass of FA-free mass of FA in the mixture)/total theoretical mass of FA) × 100(2)

### 2.9. Fourier Transform Infrared (FTIR) Spectroscopy

The infrared spectra of pure FA, pure HP-β-CD, FA/HP-β-CD-IC, gliadin fiber, G-FA fiber, and G-FA/HP-β-CD-IC fibers were investigated using a Fourier transform infrared (FTIR) spectrometer (model FTIR-8400S, Shimadzu Corp., Kyoto, Japan). The scans were done in the mid-infrared region in the range of 4000–400 cm^−1^ wavenumber at a spectral resolution of 2 cm^−1^.

### 2.10. Differential Scanning Calorimetry (DSC) and Thermogravimetric Analyze (TGA)

Thermal properties of pure FA, pure HP-β-CD, FA/HP-β-CD-IC, gliadin fiber, G-FA fiber, and G-FA/HP-β-CD-IC fibers were investigated by differential scanning calorimetry (DSC) (PerkinElmer, Akron, OH, USA) and thermogravimetric analysis (TGA) (TA Instruments, New Castle, DE, USA). The DSC analyses were conducted within a temperature range from 35 °C to 250 °C at a heating rate of 10 °C/min under N_2_ gas flow at a flow rate of 50 mL/min. The TGA measurements were carried out from 25 to 700 °C at 10 °C/min heating rate under N_2_ flow of 20 mL/min as a purge gas for both the balance and the sample.

### 2.11. X-ray Diffraction (XRD)

The crystalline structure of FA, HP-β-CD, and FA/HP-β-CD-IC was investigated using an X-ray diffractometer (model D8-ADVANCE, Bruker, Germany) with Cu Kα radiation. The samples were examined over the angular range of 2θ 5°–80°.

### 2.12. Antioxidant Activity

The antioxidant activity of free FA, G-FA, and G-FA/HP-β-CD-IC electrospun fibers was determined using the 2,2-diphenyl-1-picrylhydrazyl (DPPH) radical scavenging assay at various FA concentrations [33]. The fiber mats having the equivalent amount of FA were immersed in 2 mL of water and stirred. 2mL of 10^−4^ M DPPH solution in methanol were added to the previous solutions. The absorbance of the solutions was measured by UV-Vis spectrophotometer (UV-1280, Shimadzu Corporation, Kyoto, Japan) after 60 min. The antioxidant activity of samples was determined as:Antioxidant activity (%) = [(*A_control_* − *A_sample_*)/*A_control_*] × 100(3)
where *A_control_* and *A_sample_* are the absorbances of DPPH solution without sample and DPPH solution with the sample (Free FA or hybrid gliadin fibers), respectively.

### 2.13. Photostability

The photostability of FA incorporated within the fibers either in free form or as an inclusion complex was evaluated. Briefly, solid fibers were cut into square-shaped samples and positioned 7 cm away from a UV light source (75 W at 253.7 nm, Model NIQ 80/36 U, Heraeus, Boadilla del Monte, Madrid, Spain) in a chamber at ambient conditions. At different time intervals (0, 15, 30 and 60 min), the remaining amounts of FA were measured by UV-Vis spectroscopy at 310 nm after immersing samples into 70% ethanol solution. For comparison purposes, the free FA solution was also investigated. Each sample was analyzed at least in triplicate and the results were expressed as an average ± standard deviation.

### 2.14. In Vitro Release Assays

The in vitro release studies were carried out for selected fibers in two different media: 10% ethanol as food simulant for aqueous food products and 3% acetic acid as acidic food products [34]. In addition, we also investigated the release in PBS aqueous buffer as a biological fluid simulant. A method adapted from Atay, Fabra [35] was used for that means. Briefly, 10 mg of fibers were incorporated into 10 mL of media at ambient conditions. At specified time intervals, the samples were centrifuged at 2000 rpm for 2 min (Eppendorf centrifuge 5804r, Hamburg, Germany). Then, 1 mL aliquot of supernatant was withdrawn for analysis, replacing with fresh release medium and re-suspending. Finally, the concentration of FA in release media was calculated by measuring the absorbance of the supernatant at a wavelength of 310 nm using a UV-Vis spectrophotometer (UV-1280, Shimadzu Corporation, Kyoto, Japan). Three independent replicates of each fiber were carried out and the results were reported as average ± standard deviation.

### 2.15. Statistical Analyses

The obtained data was expressed as the mean ± standard deviation of triplicate determinations. Statistical significance among treatments were evaluated with analysis of variance (one-way ANOVA with Tukey’s post hoc test), using SPSS 25 (SPSS Inc., Chicago, IL, USA) statistical software. Tukey’s multiple range tests were applied to determine the significance of differences between mean values (*p* < 0.05).

## 3. Results and Discussion

### 3.1. Preparation of FA/HP-β-CD-IC

In recent years, CDs have gained increased attention to develop different guest-host complexes in order to improve solubility, stability, and bioavailability of a wide range of compounds [9]. A number of techniques have been developed to prepare CD-IC including co-precipitation, kneading, spray-drying and freeze-drying, among others [36]. The freeze-drying method is attracting more and more attention due to its advantages such as protection against chemical decomposition, minimal effect on guest compound activity due to processing at low temperatures as well as low moisture content amount at final physical IC [9]. Hence, we prepared FA/HP-β-CD-IC by the freeze-drying method. The content of FA in FA/HP-β-CD-IC was 11.06 ± 0.22% and the inclusion ratio of FA was 89.49 ± 0.66%. The inclusion ratio was higher than that reported by previous works [9,37], which could be ascribed to the different method of IC preparation.

### 3.2. Characterization of the Inclusion Complexes

XRD analysis has been said to be a useful technique to confirm the formation of inclusion complexes. Apparently, once the guest molecules are within the CD cavities, the crystalline peaks of guest molecules cannot be detectable [38]. Several intense and sharp diffraction peaks at 2 theta values around 9°, 10°, 16°, 17°, 27° and 29° were observed for FA (Appendix A), indicating its crystalline nature [9]. In contrast, HP-β-CD displayed an amorphous halo, confirming its amorphous character. No characteristic peaks were observed in the case of FA/HP-β-CD-IC, which could indicate the successful formation of the guest-host inclusion complexes as suggested by previous studies.

Generally, when guest molecules form inclusion complexes with cyclodextrin molecules, they might exhibit different characteristics than the pure compounds [39]. For instance, thermal transitions including melting, boiling or sublimation temperatures have been observed to shift to a different temperature or disappear [20,40]. Therefore, the DSC thermograms of FA, HP-β-CD and FA/HP-β-CD-IC were obtained. As shown in Figure 1a, the DSC thermogram of pure FA displayed a sharp endothermic peak at around 176 °C, which corresponds to its melting point, followed by a broad peak at higher temperatures attributed to decomposition. In the case of HP-β-CD, an amorphous molecule, there was a broad endothermic peak at about 84 °C, corresponding to the release of water [39]. After the formation of the inclusion complex, a different thermogram was observed. The absence of the melting point of FA and shifting of the characteristic peak of HP-β-CD (from 84 °C to 67 °C), both seemed to indicate that FA was successfully included into the cavity of HP-β-CD during the formation of the inclusion complex (Figure 1b). It has been extensively reported that when inclusion complexes with CDs are formed, the guest molecules lose their characteristic peaks in the DSC thermograms [9,37].

UV-Vis spectroscopy was also used for the characterization of the inclusion complexes. In this study, the UV-Vis spectra were recorded for FA, HP-β-CD and their inclusion complexes (Appendix A). HP-β-CD showed a very low UV absorbance without any characteristic absorption peaks. FA exhibited three characteristic peaks at 217, 287 and 310 nm, corresponding to π-π* transition of the phenyl ring, π-π* transition of the phenolic group and π-π* transition of the double bond, respectively [9]. On the other hand, in the spectrum of FA/HP-β-CD-IC, these characteristic peaks had slightly shifted (almost 2 nm to higher absorbance) suggesting the presence of non-covalent interactions between FA and HP-β-CD [41].

According to Ram, Seitz [42], FA is known to have a blue emission fluorescence wavelength at around 425 nm. Therefore, this characteristic was used to visually confirm that FA had been effectively incorporated into the HP-β-CD. As observed in Figure 2, HP-β-CD did not have any fluorescence emission while the inclusion complex with FA endowed its fluorescence properties due to the presence of FA, thus further confirming the successful IC formation.

FTIR spectroscopy was used as an additional tool to confirm the formation of a host-guest inclusion complexes. The FTIR spectra of pure FA, pure HP-β-CD, and FA/HP-β-CD-IC are depicted in Figure 3a. The FTIR spectrum of FA has characteristic peaks in the region of 3435 cm^−1^ (O–H stretching vibration), 1689, 1663, and 1618 cm^−1^ (C=O stretching vibration), 1590, 1517, and 1431 cm^−1^ (aromatic skeleton vibration). Absorption at 1466 cm^−1^ arises from single bond C–H deformations and aromatic ring vibrations while absorption at 1276 cm^−1^ is attributed the C–O–C asymmetric stretching vibration. The peak in the region of 1176 cm^−1^ is characteristic of the carbonyl group. Moreover, the bands at 852 and 804 cm^-1^ are related to the two adjacent hydrogen atoms on the phenyl ring in the FA structure [9,32]. The FTIR spectrum of HP-β-CD exhibited prominent absorption bands located at 3408 cm^−1^ (O-H stretching vibrations), 2924 cm^−1^ (C–H stretching vibrations), and 1127 and 1036 (C–H and C–O stretching vibrations) [9,43]. Several spectral changes were observed upon formation of the FA/HP-β-CD-IC, especially in the region from 1000 to 1900 cm^−1^. Comparing the HP-β-CD spectrum with that from the inclusion complex, new bands, probably arising from the incorporation of FA in the structure were observed at 1676 cm^−1^ and 1100 cm^−1^ (see arrows in Figure 3a). In addition, the characteristic peaks from the carbon stretching vibrations from the HP-β-CD shifted towards greater wavenumbers, probably due to conformational changes taking place as a consequence of incorporating the FA molecules within the CD structure.

### 3.3. Solution Properties, Fibers Morphology, and Distribution of FA Within the Fibers

In order to better understand how solution properties affected the morphology of gliadin in the presence of FA, three different concentrations of FA and FA/HP-β-CD-IC (5, 10 and 20% *w*/*v* with respect to biopolymer) were investigated. Table 1 compiles these solution properties: viscosity, electrical conductivity, and surface tension. The viscosity of G-FA and G-FA/HP-β-CD-IC solutions at all studied concentrations was higher than that of pure gliadin solutions possibly due to the interactions between the gliadin biopolymer chains and FA. On the other hand, G-FA and G-FA/HP-β-CD-IC solutions at all FA studied concentrations exhibited lower conductivity than that of the pure gliadin solution. Moreover, no significant difference was observed for the surface tension of the different solutions except for the solutions with the greatest FA concentration. 

The morphology of gliadin, G-FA and G-FA/HP-β-CD-IC electrospun fibers was investigated using scanning electron microscopy (SEM). Representative SEM images of the fiber mats and average fiber diameters (AFD) along with fiber distributions are given in Figure 4. In our previous work, the conditions for electrospinning pure gliadin fibers were optimized and uniform and beaded-free fibers having AFD 256.49 ± 78.49 nm were obtained [44]. Incorporation of free and complexed FA led to the formation of slightly thicker fibers, explained by the greater viscosity of the electrospinning solutions. Moreover, the AFD of gliadin-FA fibers was slightly higher than gliadin-FA/HP-β-CD-IC fibers. These slight variations in average fiber diameter (AFD) for fibers were most likely due to differences in solution properties such as viscosity and electrical conductivity after incorporating the bioactive compound [45,46]. As explained above, G-FA and G-FA/HP-β-CD-IC solutions at all studied concentrations had higher viscosity and lower conductivity than pure gliadin solutions. Generally, solutions with higher viscosity and lower conductivity result in thicker fibers as less stretching of the jet occurs during the electrospinning process [45,46]. In addition, the thickest fibers (279.42 ± 80.85 nm) were obtained from G-FA/HP-β-CD-IC solutions with the greatest FA content (20% *w*/*w* with respect to the polymer), which had lower conductivity and higher surface tension, thus contributing to less stretching of the electrified jet as a result of less repulsion of charges on the surface during electrospinning process [14,47]. But, in general, uniform and beaded free fibers with almost similar AFD were successfully fabricated from G-FA and G-FA/HP-β-CD-IC solutions at different FA concentrations.

Fluorescence microscopy was also used to study the distribution of FA along the fibers. As observed in Figure 2 pure gliadin fibers did not have any intrinsic fluorescence, while the hybrid fibers displayed a rather homogeneous color, suggesting that FA was effectively distributed along the fibers. It was also observed that increasing the amount of bioactive compound led to enhanced blue color intensity, confirming higher loading efficiencies at higher bioactive concentrations. The intensity of the blue color in the gliadin fibers containing the free FA was higher than in the fibers with the inclusion complexes, explained by the greater bioactive concentration in the hybrid structures.

### 3.4. Encapsulation Efficiency

The encapsulation efficiency (EE%) of FA in G-FA and G-FA/HP-β-CD-IC electrospun fibers at different concentrations were calculated by Equation 2. The data indicated that almost 100% FA was effectively incorporated into the gliadin electrospun fibers. The EE% of 5%, 10% and 20% FA-loaded fibers were 97.05 ± 0.66%, 95.09 ± 0.98% and 94.03 ± 4.88%, respectively. The EE% of 5%, 10% and 20% FA/HP-β-CD-IC-loaded fibers were 95.30 ± 1.66%, 96.93 ± 0.70% and 95.65 ± 1.38%, respectively. These values are higher than those reported for the encapsulation of FA with other electrospun biopolymers including amaranth protein isolate and pullulan [10] or using other encapsulation techniques [4,48]. Therefore, the obtained results suggest that gliadin electrospun mats can be used as an efficient encapsulant for bioactive food compounds such as FA. Given the excellent encapsulation capability of the fibers, irrespective of the added FA, the materials with the greatest amount of free FA and FA/HP-β-CD-IC (i.e., 20% with respect to the polymer) were selected for further characterization.

### 3.5. Infrared Analysis of the Electrospun Fibers

Figure 3b shows the infrared spectra of pure gliadin fibers and the hybrid electrospun fibers containing either free FA or the inclusion complexes. The spectrum of pure gliadin fibers is characterized by the bands at 1660 and 1540 cm^−1^ attributed to the C=O and C–N stretching vibration (Amide I) and N–H bending vibration and C–N and C–C stretching vibration (Amide II), respectively. Upon incorporation of free FA or the inclusion complexes, the amide I band from the fibers shifted to 1658 and 1655 cm^–1^, respectively. Similarly, a shift in the amide II band from 1540 to 1531 cm^–1^ was also observed for both hybrid fibers containing the bioactive compound. These results revealed that the incorporated FA and inclusion complexes were interacting with the amino groups from the prolamin [49]. According to Torres-Giner, Gimenez [50], the frequencies of amide I, and II reflects the size of the α-helix structure in the biopolymer; a shift toward lower wavenumbers suggests greater structural stability, which is directly related to an increase hydrogen bonding interactions taking place through the N–H groups from the protein. The band shifts observed in the FTIR spectra of G-FA and G-FA/HP-β-CD-IC electrospun fibers indicated that there were interactions among gliadin and FA or FA/HP-β-CD-IC, altering the secondary structure of the prolamin, especially when incorporating FA into gliadin electrospun fibers. In addition, there were hydrophobic interactions between the hydrophobic residue of the prolamin and phenyl ring of FA [51], which was absent in the case of incorporating FA/HP-β-CD-IC into gliadin electrospun fibers as the aromatic ring of FA in the latter case was involved in the formation of the inclusion complex.

### 3.6. Thermal Properties of the Electrospun Fibers

DSC was carried out to investigate the thermal properties of electrospun pure gliadin, G-FA, and G-FA/HP-β-CD-IC fibers. The DSC thermogram of the pure gliadin fiber exhibited a single endothermic at 80 °C, which has been normally termed as dehydration temperature (*T_d_*), corresponding the loss of bound water from the material [52]. The DSC thermograms of FA and FA/HP-β-CD-IC after incorporation into gliadin fibers were similar to gliadin fiber, showing a slight shift to around 74 °C and 73 °C, respectively; i.e., the high surface area of the generated materials facilitated water evaporation. Moreover, in the case of FA, the obtained DSC thermogram did not show the melting peak of pure FA, probably due to the previous dissolution of the bioactive for incorporation within the gliadin structures, which resulted in the crystallinity loss of the compound.

The thermal stability of FA encapsulated in the gliadin fibers in free and inclusion complex form was also investigated. The TGA studies of pure FA and gliadin fibers were also performed for comparison purposes. Appendix A summarizes the main results. In general, the weight loss in first region (less than 160 °C) is usually due to the water evaporation and volatile compounds while the temperature at which the highest rate of weight loss occurs (i.e., the peak in the derivative thermogram (DTG)) is regarded as degradation temperature (*T_d_*) [53]. Regarding the weight loss related to water evaporation (first thermal transition in the TGA curves), these could be related to the DSC curves, which explain that as very little water was weakly sorbed to the pure ferulic acid compound, the initial stability of the antioxidant molecule was greater than the other materials analyzed (refer to Figure 1). As shown in Figure 1c,d, while the first stage of water evaporation was not observed for the pure antioxidant, degradation of FA occurred in two different stages as previously observed in the literature [54], the first one corresponding to the formation of 4-vinylguaiacol (with a *T_d_* around 250 °C) and the second one mainly related to the formation of unsubstituted, 4-methyl-, and 4-ethylguaiacols at the expense of 4-vinylguaiacol [55]. After the formation of the inclusion complex with HP-β-CD, the main degradation step shifted to higher temperatures (349 °C). The enhanced thermal stability of the guest molecules within inclusion complexes has been reported for other CD-IC systems [56]. The weight loss of neat gliadin fibers took place around 320 °C. Electrospun gliadin fibers incorporating free FA exhibited a two-step degradation, at ~180 and ~310 °C, which might correspond to the *T_d_* of free FA and gliadin, respectively. This decrease in the thermal stability of the antioxidant molecule upon incorporation in protein-based matrices has been previously observed [10]. On the other hand, for G-FA/HP-β-CD-IC fibers, there was only one main degradation step with a maximum around 309 °C, indicating that the formation of inclusion complexes effectively stabilized the bioactive molecules, even though incorporation of the inclusion complexes within the electrospun gliadin fibers seemed to be somehow detrimental in terms of thermal stability (if compared with the excellent thermal stability of the isolated ICs).

### 3.7. Antioxidant Activity

The antioxidant activity of FA (powder) and after incorporation into gliadin electrospun fibers in the free and IC forms was calculated via DPPH radical scavenging assay. The antioxidant activity of FA, G-FA, and G-FA/HP-β-CD-IC fibers were 92.06 ± 1.06% 91.31 ± 0.56% and 88.79 ± 0.74% to, respectively. The results revealed that incorporation of FA within the fibers (either in free form or as an inclusion complex) did not significantly affect its antioxidant activity. FA preserved its antioxidant activity after incorporation into gliadin fibers in the free form despite the interactions that took place between FA and gliadin after electrospinning in accordance with previous studies in which similar results were obtained for other phenolic acids and prolamins [56]. In addition, the formation of the inclusion complex between FA and HP-β-CD had no effect on the antioxidant activity of FA, which might due to the high solubility of FA/HP-β-CD-IC. Aytac, Ipek [33] also reported no significant differences between the antioxidant activity of quercetin (another phenolic compound), in free form and IC form incorporated into zein fibers. Moreover, the application of high voltage during the electrospinning process had no negative effect on the antioxidant activity of FA since G-FA and G-FA/HP-β-CD-IC fibers had similar antioxidant capacity than the free FA.

### 3.8. Photostability Analyses

In general, FA might undergo photodegradation upon UV irradiation, causing trans-cis isomerization [57]. It has been reported that CDs might provide protection for their guest compounds against UV irradiation [58,59]. Hence, the photostability of the hybrid fibers was investigated. As shown in Figure 5, upon UV exposure, extensive degradation of free FA occurred, only remaining about 9% of the compound after 60 min irradiation. In contrast, incorporation within the electrospun gliadin fibers, effectively prevented its photodegradation. After 60 min of UV irradiation, the percentage of remaining FA in G-FA fibers was 43% while for G-FA/HP-β-CD-IC fibers it was 76%. In addition, the rate of FA degradation was slower for G-FA/HP-β-CD-IC fibers. It has been reported that the photostability of FA can be improved by the formation of IC with CDs including α-CD [59] and HP-β-CD [9]. In our study, the gliadin electrospun fibers provided a second shield, offering more FA stability blocking the passage of UV light towards the photosensitive compound. Therefore, FA/HP-β-CD-IC incorporated into gliadin fibers improved the photostability of FA, making it less sensitive to UV light.

### 3.9. In Vitro Release Assays

The release of FA from gliadin-FA and gliadin-FA/HP-β-CD-IC electrospun fibers in two media, 10% ethanol (as an aqueous food simulant), and 3% acetic acid (as an acidic food simulant) was studied. The obtained release profiles are depicted in Figure 6. The release of FA from G-FA and G-FA/HP-β-CD-IC electrospun fibers showed a similar behavior in acidic medium. The release of FA reached a steady state following an initial burst release that occurred during the first 10 min due to complete dissolution of both fibers in this media, which might be attributed to the greater solubility of gliadin in acetic conditions. In contrast, in the aqueous food simulant, G-FA/HP-β-CD-IC fibers quickly dissolved while swelling of the G-FA fibers occurred. Again, a fast release at the initial stage (75%) was observed for G-FA/HP-β-CD-IC fibers, while a much lower release was observed for the G-FA fibers during the first 10 min (28%), subsequently exhibiting a longer sustained release profile. The difference in the amount and rate of FA release from the fibers was probably due to the different solubility of FA in the different media. It should be highlighted that in both food simulants the release of FA from G-FA/HP-β-CD-IC fibers was greater due to the solubility enhancement in the form of an inclusion complex. It has been reported that HP-β-CD would enhance the solubility of phenolic acids including gallic acid [38]. Moreover, as explained in Section 3.5, there were hydrophobic interactions between the hydrophobic residue of the prolamin and phenyl ring of FA that were absent in the case of incorporating FA/HP-β-CD-IC into gliadin electrospun fibers as the aromatic ring of FA was involved in the formation of the inclusion complex.

In addition, the release of FA in PBS aqueous buffer as one of the most widely studied blood plasma simulant was also studied. The release FA from G-FA and G-FA/HP-β-CD-IC electrospun fibers can be divided in two stages: an initial fast release and then a slow release. The release of FA from G-FA/HP-β-CD-IC electrospun fibers was higher than G-FA fibers during the initial fast release stage. In other words, G-FA/HP-β-CD-IC electrospun fibers released almost 40% of theoretically loaded FA into PBS medium within 10 min, whereas G-FA electrospun fibers released around 27% of theoretically loaded FA over the same period. Moreover, the release of FA from G-FA/HP-β-CD-IC fibers increased gradually, reaching to a plateau (steady state) after around 7 h. In contrast, a small percentage of the theoretical FA loading was released from G-FA fibers during the same period of time. This again confirms that HP-β-CD effectively improves the solubility of FA. Additionally, HP-β-CD promoted the diffusion of water into fiber mat as a hydrating agent, increasing its porosity [60]. Thus, it could be concluded that the incorporation of FA in the form of FA/HP-β-CD-IC could be used to provide a quick solubility while requiring less amounts of FA due to the high solubility of FA in this case. On the contrary, G-FA fiber mats might serve as matrices for more sustained release applications.

## 4. Conclusions

In this work, FA was successfully incorporated into gliadin fibers in form of FA/HP-β-CD-IC via the electrospinning technique. First of all, the inclusion complex between FA and HP-β-CD was prepared at a 1:1 molar ratio using a freeze-drying method. The formation of the inclusion complex between FA and HP-β-CD was confirmed by FTIR, DSC, and XRD analyses. Then, FA/HP-β-CD-IC was incorporated into gliadin fibers in order to fabricate hybrid electrospun fibers. Gliadin fibers incorporating FA in free form were also electrospun for comparative purposes. The uniform and beaded-free fibers and FA distribution along the fibers were observed using SEM and fluorescence microscopy, respectively. The obtained electrospun fibers maintained their antioxidant activities in spite of the high voltage applied during the electrospinning process. Moreover, the photostability of FA was significantly improved when incorporating the bioactive in the form of FA/HP-β-CD-IC. The inclusion complexes also favored the solubility of FA in different media. Hence, these fibers could find certain applications in various areas including food, packaging, health, pharmaceutical, among others.

## Figures and Tables

**Figure 1 nanomaterials-08-00919-f001:**
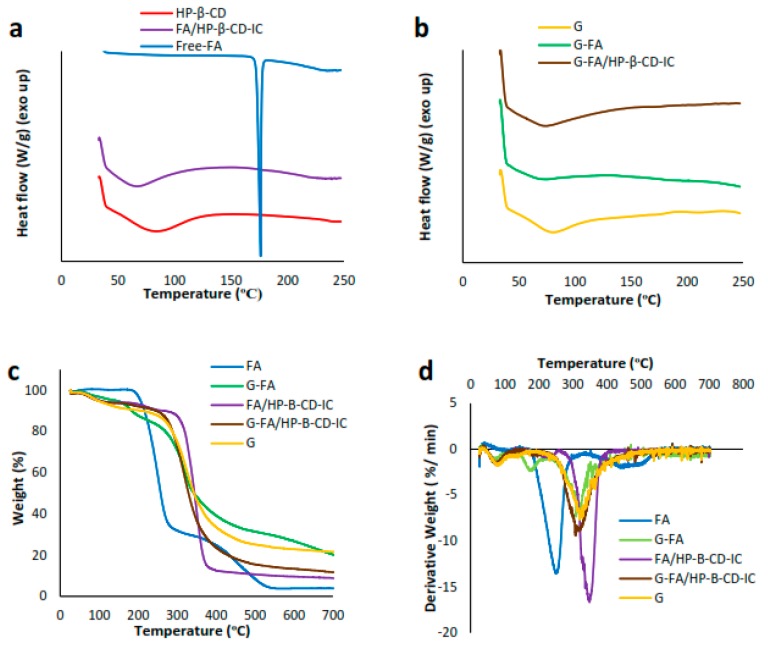
(**a**,**b**) differential scanning calorimeter (DSC) thermograms of pure ferulic acid (FA), HP-β-CD and FA/HP-β-CD-IC and Gliadin, G-FA 20%, and G-FA/HP-β-CD-IC 20% electrospun fibers, respectively. (**c**) Thermogravimetric analysis (TGA) and (**d**) Derivative thermogravimetric (DTG) curves of pure FA, and FA/HP-β-CD-IC, Gliadin, G-FA 20%, and G-FA/HP-β-CD-IC 20% electrospun fibers.

**Figure 2 nanomaterials-08-00919-f002:**
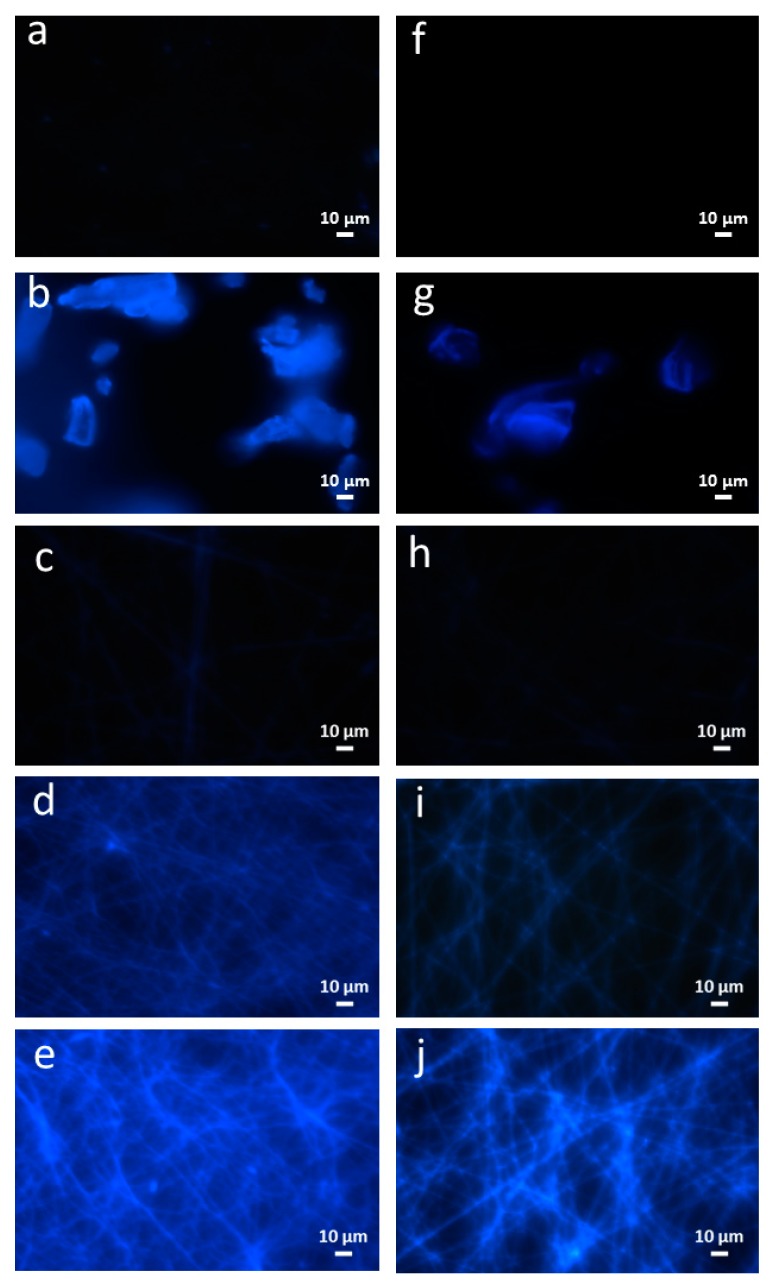
Fluorescence microscopy images of electrospun structures: (**a**) pure gliadin; (**b**) pure FA; (**c**) G-FA 5%; (**d**) G-FA 10%; (**e**) G-FA 20%; (**f**) pure HP-β-CD; (**g**) FA/HP-β-CD-IC; (**h**) G-FA/HP-β-CD-IC 5%; (**i**) G-FA/HP-β-CD-IC 10%; (**j**) G-FA/HP-β-CD-IC 20%.

**Figure 3 nanomaterials-08-00919-f003:**
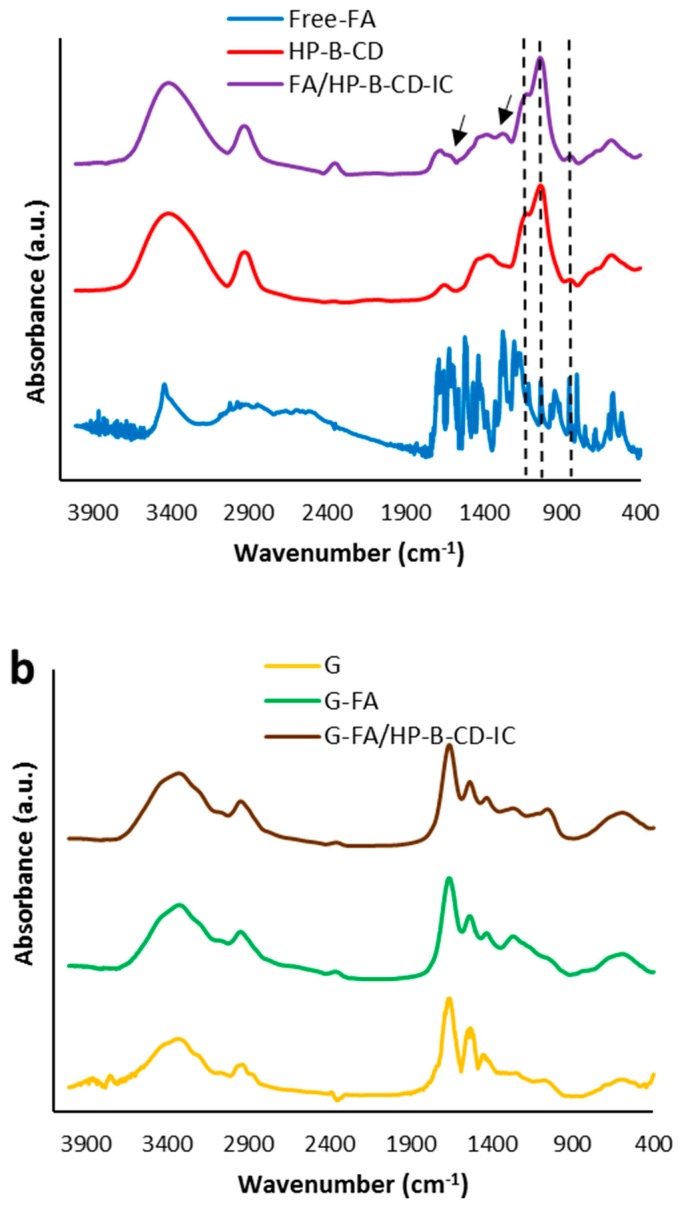
Fourier transform infrared (FTIR) spectra of (**a**) pure FA, HP-β-CD and FA/HP-β-CD-IC; (**b**) Gliadin, G-FA 20%, and G-FA/HP-β-CD-IC 20% electrospun fibers.

**Figure 4 nanomaterials-08-00919-f004:**
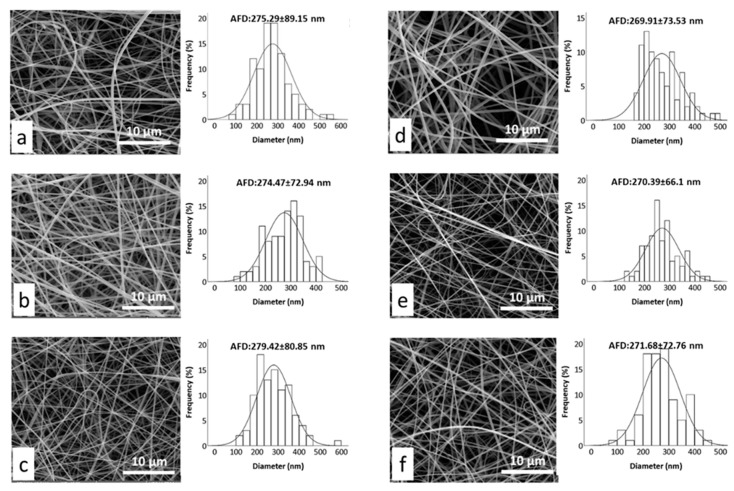
Representative scanning electron microscope (SEM) images and average fiber diameter (AFD) of electrospun structures: (**a**) G-FA 5%; (**b**) G-FA 10%; (**c**) G-FA 20%; (**d**) G-FA/HP-β-CD-IC 5%; (**e**) G-FA/HP-β-CD-IC 10%; (**f**) G-FA/HP-β-CD-IC 20%.

**Figure 5 nanomaterials-08-00919-f005:**
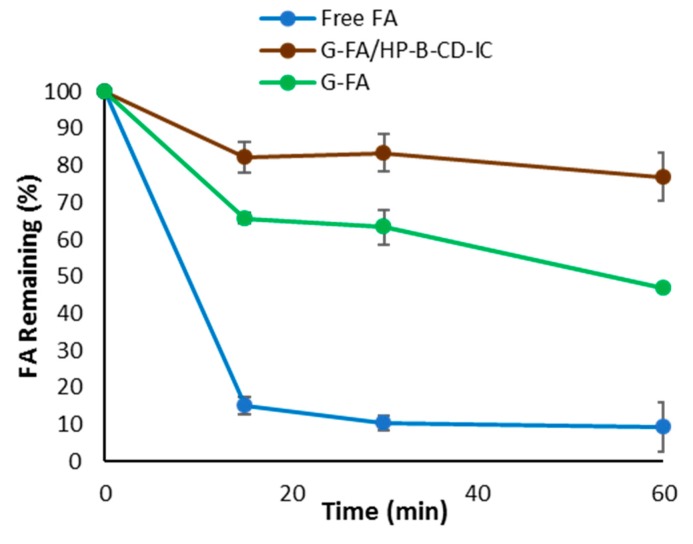
Photodegradation profiles of pure FA, G-FA 20% and G-FA/HP-β-CD-IC 20% electrospun fibers.

**Figure 6 nanomaterials-08-00919-f006:**
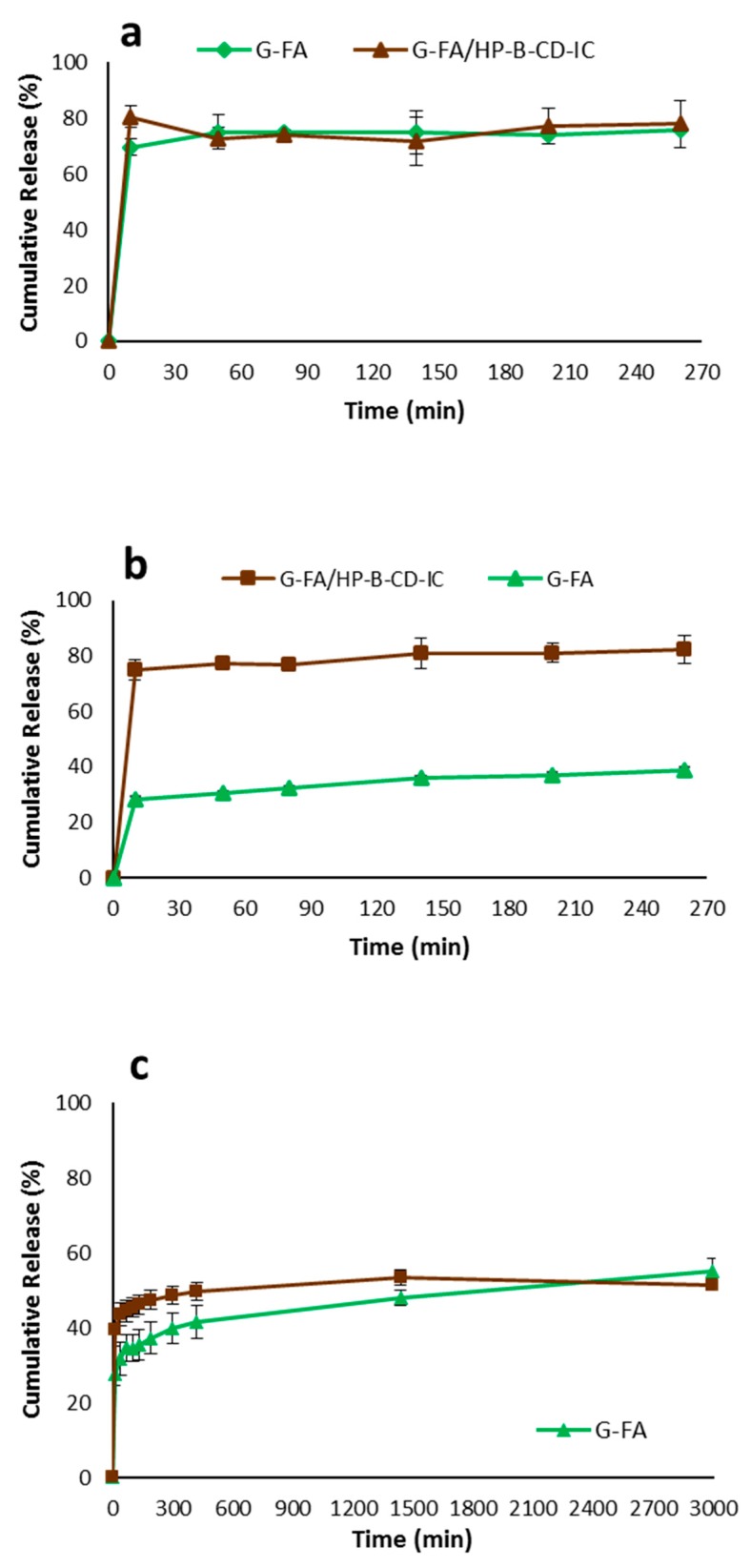
Release behavior of FA from G-FA 20% and G-FA/HP-β-CD-IC 20% electrospun fibers in different media: (**a**) acetic acid 3%; (**b**) ethanol 10%; and (**c**) PBS.

**Table 1 nanomaterials-08-00919-t001:** Solution concentrations and properties used in electrospinning process.

Solution	%Gliadin ^1^ (*w*/*v*)	%HP-β-CD ^2^ (*w*/*w*)	%Ferulic acid ^2^ (*w*/*w*)	%Ferulic acid/HP-β-CD IC ^2^ (*w*/*w*)	Viscosity ^3^ (mPa.s 10 rpm, 25 °C)	Surface Tension ^3^ (mN.m^−1^)	Electrical Conductivity ^3^ (μs.cm^−1^)
Gliadin	25	-	-	-	197.9 ± 1.61 ^e^	28.80 ± 0.37 ^b^	148.25 ± 0.78 ^a^
Gliadin/Ferulic acid	25	-	5	-	219.1 ± 2.71 ^d^	29.05 ± 0.08 ^ab^	89.70 ± 1.73 ^bc^
	25	-	10	-	238.3 ± 2.44 ^bc^	29.30 ^ab^	90.17 ± 0.93 ^bc^
	25	-	20	-	247.2 ± 3.92 ^b^	29.80 ± 0.14 ^a^	81.1 ± 0.71 ^d^
Gliadin/Ferulic acid/HP-β-CD IC	25	-	-	5	219.1 ± 7.56 ^d^	29.35 ± 0.35 ^ab^	93 ± 0.85 ^b^
	25	-	-	10	244.4 ± 7.37 ^b^	29.55 ± 0.08 ^ab^	87.50 ± 3.39 ^c^
	25	-	-	20	277.1 ± 9.43 ^a^	29.45 ± 0.35 ^ab^	89.03 ± 1.58 ^bc^

1—With respect to the solvent (acetic acid). 2—With respect to the polymer (gliadin). 3—Data are displayed in means ± standard deviation of three replications (*p* < 0.05); means in each column bearing different superscripts are significantly different (*p* < 0.05).

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
