# Peer review of "Active Food Packaging Coatings Based on Hybrid Electrospun Gliadin Nanofibers Containing Ferulic Acid/Hydroxypropyl-Beta-Cyclodextrin Inclusion Complexes"

_nanomaterials, 2018, doi:10.3390/nano8110919_

Reviewer 1 Report

The paper deals with the preparation and characterization of active food packaging formed by electrospun fibers containing inclusion complexes of ferulic acid (FA) with hydroxypropyl-beta-cyclodextrins. The topic is interesting and the manuscript is well written and organized. I suggest the publication in Nanaomterials after the following revisions:

-          The Introduction section should be improved by highlighting that the combination of ecocompatible materials (such as biopolymers and clay nanopartocles containing functional molecules) represents an effective strategy to obtain active biocompatible packaging. Within this, recent articles on functional nanocomposites based on pectins (Carbohydrate Polymers 2017, 170, 198–205), polylactic acid (Molecules 2017, 22(7), 1170), chitosan (New J. Chem., 2018,42, 8384-8390), and cellulose (Green Materials 2014, 2, 232-242) should be quoted.

-          TGA measurements. The Nitrogen flows for both the balance and the sample should be indicated.

-          A more detailed presentation and discussion of the TGA results is necessary. I suggest to define the different degradation steps of the investigated materials (the degradation temperature should be estimated by the maxima of the DTG curves). Additionally, the residual matter at 700 °C should be determined as well as the mass losses up to 150 °C. These parameters should be presented in an additional table.

-          Figure 2c.  y-axis should be indicated as “Weight / %” being that the first value is 100.

-          Figure 2d. y-axis should be indicated as “Derivative Weight / % min-1”

-       Figure 6. The y-axis represents the cumulative release. Therefore, the release profile should present continuously increasing trends.

Author Response

The paper deals with the preparation and characterization of active food packaging formed by electrospun fibers containing inclusion complexes of ferulic acid (FA) with hydroxypropyl-beta-cyclodextrins. The topic is interesting and the manuscript is well written and organized. I suggest the publication in Nanaomterials after the following revisions:

-          The Introduction section should be improved by highlighting that the combination of ecocompatible materials (such as biopolymers and clay nanoparticles containing functional molecules) represents an effective strategy to obtain active biocompatible packaging. Within this, recent articles on functional nanocomposites based on pectins (Carbohydrate Polymers 2017, 170, 198–205), polylactic acid (Molecules 2017, 22(7), 1170), chitosan (New J. Chem., 2018,42, 8384-8390), and cellulose (Green Materials 2014, 2, 232-242) should be quoted.

As requested by the reviewer, the information about the works related to combining biopolymers with nanoclays and functional molecules has been included in the introduction with some of the suggested references from the reviewers.

-          TGA measurements. The Nitrogen flows for both the balance and the sample should be indicated.

Thank you for the comment. The information has been included as requested.

-          A more detailed presentation and discussion of the TGA results is necessary. I suggest to define the different degradation steps of the investigated materials (the degradation temperature should be estimated by the maxima of the DTG curves). Additionally, the residual matter at 700 °C should be determined as well as the mass losses up to 150 °C. These parameters should be presented in an additional table.

As requested by the reviewer, a more thorough analysis of the TGA results has been included as well as a Table (as supplementary material) with the more relevant thermal values.

-          Figure 2c.  y-axis should be indicated as “Weight / %” being that the first value is 100.

Thank you for the comment. We assume that the reviewer refers to Figure 1c, which has been changed as requested.

-          Figure 2d. y-axis should be indicated as “Derivative Weight / % min-1”

Thank you for the comment. The y-axis has been modified as requested.

-       Figure 6. The y-axis represents the cumulative release. Therefore, the release profile should present continuously increasing trends.

We agree with the reviewer in that one should expect a continuous increase in release. However, from the results it can be observed that a quick release occurs during the first stages, while there is a fraction of the bioactive that must be strongly interacting with the encapsulation matrix that it is not released with time and, thus, the amount of bioactive released remains constant after a certain time period.

Reviewer 2 Report

The work presents some new interesting results and should be revised before publication for clarity and in order to improve data analysis.      

     -    Introduction. I recommend to quote recent reviews (Appl. Sci. 2018, 8(7), 1068; ) and articles (Applied Clay Science, 160, 2018, 95-105; Materials 2018, 11(4), 575; ACS Appl. Mater. Interfaces, 2017, 9, 17476–17488; Nanomaterials 2018, 8(10), 793) on the development of ecocompatible hybrid  materials suitable as active food packaging      
     -    DSC analyses. Did the authors perform two heating/cooling cycles on the same sample to study the “history” of the investigated materials?      
     -    DSC results. Based on the enthalpy of water evaporation, the water content of all the materials could be calculated by considering the endothermic peak at 80-100 °C. Details on this calculation are reported in Langmuir 2011, 27, 1158–1167. The obtained values should be compared to those estimated from the weight losses up to 150 °C (TG curves).      
     -    SEM analyses. The working distance should be reported in the Experimental section.      
     -    A fitting analysis on the release data (Figure 6) is recommended

Author Response

The work presents some new interesting results and should be revised before publication for clarity and in order to improve data analysis.      

     -    Introduction. I recommend to quote recent reviews (Appl. Sci. 2018, 8(7), 1068; ) and articles (Applied Clay Science, 160, 2018, 95-105; Materials 2018, 11(4), 575; ACS Appl. Mater. Interfaces, 2017, 9, 17476–17488; Nanomaterials 2018, 8(10), 793) on the development of ecocompatible hybrid materials suitable as active food packaging.

As requested by the reviewer, some of the references suggested by the reviewers have been included in the introduction on the development of ecocompatible hybrid materials for active food packaging.

     -    DSC analyses. Did the authors perform two heating/cooling cycles on the same sample to study the “history” of the investigated materials?

We just analysed the first heating and cooling run, as we were interested in the properties of the as-prepared materials and not after erasing the thermal history.

     -    DSC results. Based on the enthalpy of water evaporation, the water content of all the materials could be calculated by considering the endothermic peak at 80-100 °C. Details on this calculation are reported in Langmuir 2011, 27, 1158–1167. The obtained values should be compared to those estimated from the weight losses up to 150 °C (TG curves).

Thank you for the comment. Although we consider this Langmuir study very interesting, the information we could get from this analysis is not that relevant for the current study. However, we have included some information relating both techniques which can explain why ferulic acid seems more thermally stable in this first degradation step.

     -    SEM analyses. The working distance should be reported in the Experimental section.

Working distances have been included as requested.

     -    A fitting analysis on the release data (Figure 6) is recommended.

Fitting analysis was attempted but the results obtained were not reliable as there were few release points when the actual release was taking place (i.e. during the first time points).

Reviewer 3 Report

Nice work and good presentation.

Author Response

Nice work and good presentation.

Thanks for the comments

Reviewer 4 Report

Dear Authors, 

I read with great interest your manuscript and I found it very well presented and with a great scientific soundness.

In my opinion it is suitable for publication after minor revision.

My comments:

Abstract: 

Line 24-27: adjust the sentence

Abstract can be improved.

Introduction

Line 36, page2: what mean "IC" in the FA/HP-β-CD-IC ? Please add the explanation of the acronym.

Experimental part:

- Line 8, page 3: add the Pa value.

- After the 2.2 session, add a Table with the sample prepared.

- Line 13, page 3: 10000 g is correct?

-Line 40, page 3: .... dissolved in methanol  ....

Figure 1, supplementary materials: adjust the sample indication (red line) as HP-B-CD-IC like Figure 2.

Results and discussion:

- Line 23 page 6: double word "into included"

Author Response

Dear Authors, 

I read with great interest your manuscript and I found it very well presented and with a great scientific soundness.

In my opinion it is suitable for publication after minor revision.

My comments:

Abstract: 

Line 24-27: adjust the sentence

Adjusted as requested.

Abstract can be improved.

We have tried to improve the abstract as requested.

Introduction

Line 36, page2: what mean "IC" in the FA/HP-β-CD-IC ? Please add the explanation of the acronym.

The full term has been included before the acronym.

Experimental part:

- Line 8, page 3: add the Pa value.

We are not sure what the referee refers to with “Pa value”.

- After the 2.2 session, add a Table with the sample prepared.

Thanks for the suggestion. However, as only one type of inclusion complex was prepared we consider that it is not necessary to add a table.

- Line 13, page 3: 10000 g is correct?

Yes, it is correct.

-Line 40, page 3: .... dissolved in methanol  ....

It has been corrected.

Figure 1, supplementary materials: adjust the sample indication (red line) as HP-B-CD-IC like Figure 2.

The sample name has been changed as requested.

Results and discussion:

- Line 23 page 6: double word "into included"

Thanks for the observation. The sentence has been corrected.

Round  2

Reviewer 1 Report

The paper can be accepted in the current form.

Reviewer 2 Report

revised version is suitable for publication